# The acute effects of knee extension exercises with different contraction durations on the subsequent maximal knee extension torque among athletes with different strength levels

**Kaito Nakata**[1]*, **Takaaki Mishima**[1,2]

**1** Graduate School of Sport and Exercise Sciences, Osaka University of Health and Sport Sciences, Sennan, Osaka, Japan, **2** Osaka University of Health and Sport Sciences, School of Health and Sport Sciences, Sennan, Osaka, Japan

* nakakai.722@gmail.com

**Data Availability Statement:** The data underlying the results presented in the study are available

## Abstract

Individuals with high fatigue resistance against a high-intensity conditioning activity (CA) may be able to avoid experiencing significant fatigue and enhance their voluntary performance. We examined whether the optimal contraction duration of dynamic knee extension exercises to maximize subsequent voluntary performance varies depending on the strength level of an individual. The study participants were 22 male American college football players. Initially, all participants performed a 10-s maximal isometric knee extension exercise and were classified as stronger individuals (n = 8) and weaker individuals (n = 8) based on their relative muscle strength. Each group then performed three types of dynamic CA with different contraction durations (6 s [6-CA], 12 s [12-CA], and 18 s [18-CA]) in random order. To observe the time-course changes in post-activation potentiation and performance enhancement, the twitch torques induced by electrical stimulation and isokinetic knee extension torques at 180˚/s were recorded before and after each CA. The twitch torque increased at 10 s (29.5% ± 9.3%) and 1 min (18.5% ± 6.8%) after 6-CA for the stronger individuals ($p < 0.05$). However, no post-activation potentiation was induced in the weaker individuals in either protocol. Voluntary performance increased at 4 (7.0% ± 4.5%) and 7 (8.2% ± 4.3%) min after 18-CA for stronger individuals ($p < 0.05$). However, there was no post-activation performance enhancement in either protocol for weaker individuals. Thus, CA with a relatively long contraction duration was optimal to maximize the subsequent voluntary performance for stronger individuals. It remains unknown whether CAs performed with relatively short or long contraction durations were optimal for weaker individuals.

## Introduction

The twitch torque evoked by a single electrical stimulus is increased transiently after a maximal or submaximal conditioning activity (CA) [1, 2]. This phenomenon is referred to as post-

**Funding:** The authors received no specific funding
for this work.

**Competing interests:** The authors have declared
that no competing interests exist.

activation potentiation (PAP). The primary mechanism of PAP is the phosphorylation of myosin regulatory light chains during a CA, which transiently increases the sensitivity of actin–myosin interactions to a given amount of $Ca^{2+}$ [3, 4]. As a result, the force-generating capability of the muscle is enhanced [5, 6]. Several previous studies have reported that a CA can improve not only the twitch torque evoked by electrical stimulation but also the subsequent voluntary performance [7–11], which is called the post-activation performance enhancement (PAPE) [12]. Tillin and Bishop [6] described the phosphorylation of myosin regulatory light chains as one of the major mechanisms of PAPE. In practice, coaches can utilize PAPE through high-intensity resistance exercises as CA to enhance the performance of subsequent plyometric exercises (i.e., complex training). Alternately, they can help enhance performance by employing CA immediately prior to the actual competitive performance (e.g., 100-m performance).

The potentiating effect and fatigue associated with a CA coexist [13]. Therefore, an increase in voluntary performance after a CA appears to depend on the net balance between fatigue and potentiation [6]. Specifically, voluntary performance may decrease if the effects of fatigue are dominant but increase if those of potentiation dominate or may remain unchanged if the effects of fatigue and potentiation are at similar levels. Therefore, in situations where coaches are attempting to employ a CA to sufficiently enhance the subsequent voluntary performance (i.e., complex training or prior to the actual competition), they need to consider both inducing the potentiating effect and suppressing the fatigue associated with a CA as much as possible. It has been reported that the contraction duration of a CA plays a crucial role in maximizing PAPE [6, 14, 15]. As fatigue and potentiation through CA are closely intertwined, ways to optimize total contraction duration need to be considered for maximizing PAPE. If the total contraction duration of a CA is too short, the mechanisms associated with the potentiating effect may not be triggered [14]. Seitz et al. [14] examined the effect of the total contraction duration of isokinetic dynamic knee extension exercises on PAP and PAPE. They observed that a minimum total contraction duration must be reached during a CA to trigger the mechanisms responsible for PAP and PAPE. However, if the total contraction duration of a CA is too long, the potentiating effect induced by the CA may be counteracted by significant fatigue [15]. It follows that to effectively induce PAPE, it is necessary to identify the optimal total contraction duration of a CA that can sufficiently trigger the potentiating effect while minimizing the fatigue as much as possible.

Interestingly, previous studies have suggested that the magnitude of PAP [1] and fatigue [16–18] after a high-intensity CA vary depending on the strength level of an individual. Specifically, compared to individuals with relatively lower muscle strength, those with a relatively higher muscle strength may induce higher PAP due to a higher proportion of fast-twitch fibers [19, 20], which determines the degree of PAP [1]. Such individuals may also exhibit higher fatigue resistance to high-intensity CA [16–18]. Given that the magnitude of PAP and fatigue after a high-intensity CA vary depending on the strength level of an individual, we hypothesized that the optimal contraction duration of a CA to maximize PAPE may differ between stronger and weaker individuals. If a high-intensity CA with a relatively long contraction duration (i.e., approximately 20 s) is performed, it is likely that PAPE is not achieved because significant fatigue may counteract the potentiating effect. However, stronger individuals with high fatigue resistance against a high-intensity CA may have the potential to suppress the development of significant fatigue and enhance the subsequent voluntary performance, even in a CA with a relatively long total contraction duration. Furthermore, in stronger individuals with a higher number of fast-twitch fibers, the PAP mechanism may be triggered to a great extent by the performance of CA with a long total contraction duration, resulting in a higher level of PAPE. On the other hand, when a high-intensity CA with a relatively short contraction

duration (i.e., approximately 5 s) is performed, the magnitude of the potentiating effect is expected to be lower than that observed when a high-volume CA is performed. However, a short contraction duration may produce less fatigue and allow the potentiating effect to dominate, even in weaker individuals with relatively low fatigue resistance. Thus, the optimal contraction duration of a CA to maximize PAPE may vary depending on the strength level of an individual. A recent meta-analysis reported that stronger individuals tend to show more PAPE when a high-intensity CA is performed at maximal repetition than at sub-maximal repetition, whereas the opposite may be observed for weaker individuals [21]. However, to the best of our knowledge, no studies have examined in detail whether the optimal contraction duration of a high-intensity CA differs among individuals with different muscle strength levels.

Therefore, this study aimed to verify whether the optimal contraction duration of a high-intensity CA to maximize PAPE varies depending on the strength level of an individual. We hypothesized that CA with a relatively long contraction duration (i.e., approximately 20 s) would be optimal for stronger individuals, while the CA with a relatively short contraction duration (i.e., approximately 5 s) would be optimal for weaker individuals. This hypothesis, if proven, will reaffirm the importance of individualizing strategies based on the level of muscle strength.

## Materials and methods

### Participants

Twenty-two male American college football players participated in this study (Table 1). All participants had performed basic resistance exercises, including high-intensity bench press and squats in a session of training, approximately twice a week for at least six months. In addition, the present study was conducted in the pre-season period, and power exercises such as power cleans were also introduced into the training program. During the experimental period, they were instructed to continue their daily routines and not change their physical activity level and food and fluid intakes. Moreover, they were asked to avoid consuming depressive (e.g., alcohol) or ergogenic (e.g., coffee) substances 24 h prior to the experimental sessions. They had been free from musculoskeletal injuries for at least one year before the study. The participants were informed verbally and in writing about the procedures involved, and possible risks and benefits of the study, for which written informed consent was obtained from all study participants before the study commenced. This study was approved by the Ethics Review

**Table 1. Physical characteristics, MVC, twitch torque, and %PAPMVC in the whole group and stronger and weaker individual subgroups.**

| | Whole (n = 22) | Stronger individuals (n = 8) | Weaker individuals (n = 8) |
|---|---|---|---|
| Age (years) | 19.7 ± 1.2 | 19.9 ± 1.6 | 19.8 ± 1.0 |
| Training Age (years) | 3.5 ± 1.2 | 3.5 ± 1.1 | 3.5 ± 1.7 |
| Height (m) | 1.72 ± 0.06 | 1.73 ± 0.06 | 1.69 ± 0.06 |
| Body mass (kg) | 74.6 ± 10.6 | 72.3 ± 10.4 | 74.8 ± 13.2 |
| MVC (N·m) | 274.7 ± 68.2 | 337.6 ± 41.6* | 204.3 ± 37.3 |
| MVC/Body mass (N·m·kg$^{-1}$) | 3.7 ± 0.9 | 4.7 ± 0.5* | 2.8 ± 0.5 |
| pre-MVC (N·m) | 44.6 ± 8.9 | 43.7 ± 6.0 | 47.3 ± 9.4 |
| post-MVC (N·m) | 71.3 ± 10.6 | 75.3 ± 8.1 | 67.1 ± 12.9 |
| %PAP$_{MVC}$ (%) | 63.2 ± 29.1 | 73.5 ± 11.2* | 43.4 ± 22.4 |

Values are presented as means ± standard deviation. MVC, maximal voluntary contraction; pre-MVC, twitch torque before MVC; post-MVC, twitch torque after MVC. The statistical difference is set at $p < 0.05$: difference versus weaker individuals*.

Committee of Osaka University of Health and Sport Sciences (approval number: 21–2). The study was conducted in conformity with the policy statement regarding the use of human participants by the Declaration of Helsinki. With the permission of the Ethics Committee, this study did not require parental consent for the participation of minors as the experimental interventions were minimally invasive.

## Study design

This study was conducted over four sessions for each participant. In the first session, the participants were grouped into stronger individuals or weaker individuals based on their relative muscle strength, which was calculated by dividing the isometric maximal knee extension torque by the body mass. As body size affects the tested muscle strength, Jaric et al. [22] recommend normalization method that divides torque according to body mass. For the latter three sessions (i.e., experimental sessions), the participants who were classified as stronger or weaker individuals in the first session were asked to participate. Each experimental session was conducted at least 24 h apart. All sessions were conducted between 10:00 AM and 3:00 PM in an indoor laboratory. To determine the optimal contraction duration of a CA to maximize PAPE, each participant performed three types of dynamic CAs with different total contraction durations (6 s [6-CA], 12 s [12-CA], and 18 s [18-CA]) in random order. Isokinetic knee extension exercises were performed for dynamic CA, and the joint angular velocity of each CA was set at 60˚/s. To observe the time-course changes in PAP and PAPE, before and after each CA, the twitch torques induced by electrical stimulation and voluntary knee extension torques at 180˚/s were recorded. In all sessions, the participants were required to limit their activities such as walking for 10 min after arriving at the laboratory and remaining at rest to eliminate the effect of PAP induced while traveling to the laboratory. Given that PAP declines quasi-exponentially and is negligible after about 5 minutes [1, 2], this rest period is sufficient to dissipate the effects of PAP. In addition, Macintosh et al. [23] indicated a possibility that the learning and familiarization effect of the performance task could affect PAPE. Therefore, in the present study, we conducted a task-specific warm-up session, based on the study by Seitz et al. [14], before each experimental session to eliminate the possibility of the learning and familiarization effect.

## Procedure

**Classification of stronger and weaker individuals.** To classify stronger and weaker individuals, all participants performed a 10-s maximal voluntary isometric knee extension exercise. The participants sat on a dynamometer (Biodex System 4; Sakai Medical Instrument, Tokyo, Japan) and were tightly secured to the seat using two crossover seatbelts and a waist harness. The lever arm of the dynamometer was attached 2–3 cm above the lateral malleolus with a strap. The rotation axis of the knee was aligned with the axis of the motor. Based on a previous study by Miyamoto et al. [24] the knee and hip joint flexion angles were fixed at 90˚ and 80˚, respectively. The participants were consistently encouraged to contract "as forcefully and as fast as possible". This instruction is considered favorable for measuring maximal force and power [25, 26]. To observe the relationship between the muscle strength levels and the magnitude of PAP, a twitch torque was induced by electrical stimulation before and immediately after a 10-s isometric knee extension exercise. The extent of PAP induced by isometric knee extension exercise (%PAP$_{MVC}$) was calculated as follows:

$$\%PAP_{MVC} = \frac{(T_{twpost-MVC} - T_{twpre-MVC})}{T_{twpre-MVC}} \times 100$$

where $T_{twpre-MVC}$ and $T_{twpost-MVC}$ are the maximal twitch torques obtained before and

immediately after the isometric knee extension exercise, respectively. From an initial sample of 22 men, 8 participants demonstrating the highest and 8 demonstrating the lowest relative muscle strength were selected and classified as stronger and weaker individuals, respectively. The other 6 participants were not classified into any of the groups and did not participate in any further experiments. The G*Power software version 3.1 was used to determine the sample size of the stronger or weaker group. The following parameters were selected: medium effect size (f = 0.25), an alpha level of .05, a power level of 0.8. The sample size for both the stronger and weaker groups was determined to be at least 8. The peak torque of voluntary knee extension and twitch contraction of the dynamometer was analog-to-digital converted (Power Lab/8SP; ADInstruments, Bella Vista, NSW, Australia) and stored on personal computer software (Lab-Chart 6 Japanese; ADInstruments). Data were filtered by a low-pass Butterworth filter with a cut-off frequency of 12 Hz. This cutoff frequency was determined based on visual inspection in our preliminary experiments. Specifically, it was determined by confirming that 3 standard deviation of the noise was no greater than the threshold of 0.5 N for the contraction onset of isometric knee extension [27]. The sampling frequency was set at 100 Hz.

**Electrical stimulation procedure.** To evoke quadriceps muscle twitches, an electrode pad consisting of a $10 \times 20$ cm aluminum foil with an adhesive conductor (SR-4080; Minato Medical Science, Osaka, Japan) attached and covered with kitchen paper was prepared. Two electrode pads were wetted with water and attached to the upper and lower anterior thigh to cover the entire quadriceps muscle of the dominant leg. The participant was then placed on the dynamometer seat with the knee and hip joints fixed at 90˚ and 80˚, respectively. The electrode pads were connected to a high-voltage constant-current stimulator (Model DS7AH; Digitimer Ltd, Hertfordshire, UK) via an output cable (D185-HB4; Digitimer Ltd, Hertfordshire, UK). The stimulus intensity was set at 120% of the maximal intensity [15, 28].

**Task-specific warm-up sessions.** An overview of the task-specific warm-up session is shown in Fig 1A. After we determined the stimulation intensity, the participants performed two isokinetic knee extension exercises at 180˚/s at 20%, 40%, 60%, and 80% of their perceived maximal torque at 45-s intervals. Isokinetic knee extensions at 100% of the maximal torque were then performed "as fast and as hard as possible" every minute until the peak torque production in three consecutive contractions differed by less than 2%. According to a previous study [14], the angular velocity of the performance test was set at 180˚/s. In addition, the range of motion of the knee was between 110˚ and 20˚ (0˚, full extension) in concentric contractions. The participant performed a maximal isokinetic knee extension exercise over the entire range of motion until the dynamometer mechanically stopped.

**Experimental session.** After the task-specific warm-up session, a 10-min rest period was provided before the experimental session. An overview of the experimental session and the post-test protocol are shown in Fig 1B and 1C, respectively. In the experimental session, the responses to one isometric twitch torque ($Pre_{before}$) and two isokinetic knee extension exercises at 180˚/s followed 5-s later by one isometric twitch ($Pre_{after}$) were recorded before each participant performed each CA. The last stimulation was used to examine the possible potentiating effect of a set of two maximal isokinetic knee extensions at 180˚/s. Following a rest period of approximately 90-s after $Pre_{after}$, one of the three dynamic CAs with different contraction durations (6-CA, 12-CA, or 18-CA) was performed. The post-test was completed 10 s (Post-10 s) and 1 (Post-1 min), 4 (Post-4 min), 7 (Post-7 min), and 10 (Post-10 min) min after each CA (Fig 1B). The post-test protocol consisted of a single stimulation and two isokinetic knee extension exercises at 180˚/s (Fig 1C). The range of motion of the knee was the same as in the task-specific warm-up session. The knee extension resulting in the highest voluntary peak torque at each time point (i.e., Pre, Post-10 s, Post-1 min, Post-4 min, Post-7 min, and Post-10 min) was selected for further analysis. After the first knee extension exercise, the lever arm of

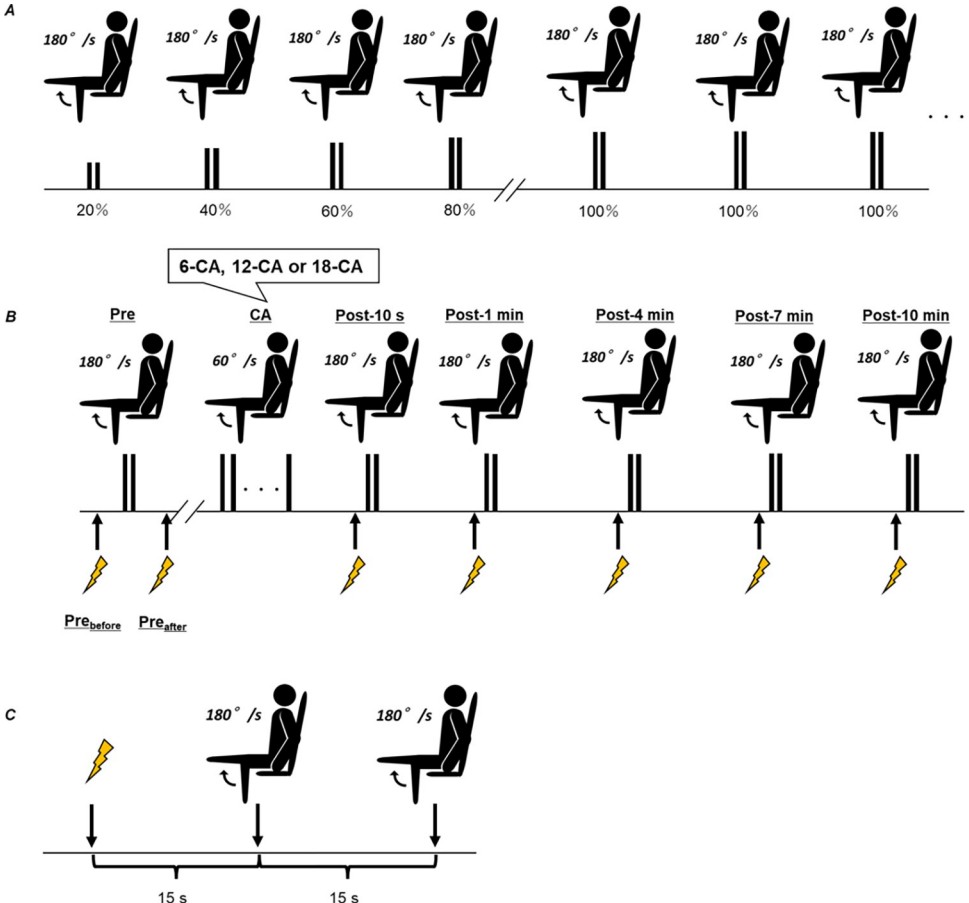

**Fig 1. Experimental overview.** (*A*) Task-specific warm-up procedure (*B*) Experimental procedure for the 6-CA, 12-CA, and 18-CA protocols. (*C*) The details of the test protocol after each CA. CA, conditioning activity.

the dynamometer was manually returned to the starting position (i.e., 110˚ of knee flexion) and then the second knee extension exercise was performed. The extent of PAPE (%PAPE) was calculated as follows:

$$\%PAPE = \{(T_{vol.post-CA} - T_{vol.pre-CA})/T_{vol.pre-CA}\} \times 100$$

where $T_{vol.pre-CA}$ and $T_{vol.post-CA}$ are the maximal knee extension torques during isokinetic knee extension exercises before and after each CA, respectively. The extent of PAP measured in the experimental session was expressed as %PAP. Among them, the %PAP observed in two maximal knee extension exercises during the pre-test was calculated as follows:

$$\%PAP = \{(T_{tw.preafter-CA} - T_{tw.prebefore-CA})/T_{tw.prebefore-CA}\} \times 100$$

where $T_{tw.prebefore-CA}$ and $T_{tw.preafter-CA}$ are the maximal twitch torques induced before and after the two isokinetic knee extension exercises during the pretest, respectively. Finally, the %PAP observed after each CA was calculated as follows:

$$\%PAP = \{(T_{tw.post-CA} - T_{tw.prebefore-CA})/T_{tw.prebefore-CA}\} \times 100$$

where $T_{tw.post-CA}$ is the maximal twitch torque induced at each time point after each CA. The repeatability (intraclass correlation coefficient [ICC]) of the peak torque of both maximal

voluntary isokinetic knee extension and twitch contraction was investigated in our preliminary study. The ICC values were 0.962 (95% confidence interval (CI): 0.897–0.989) and 0.998 (95% CI: 0.995–1.000) for voluntary knee extension peak torque and twitch peak torque, respectively.

**CA protocols.** Based on a previous study by Seitz et al. [14], the dynamic CAs employed in this study were isokinetic knee extension exercises at 60˚/s. The numbers of contractions for 6-CA, 12-CA, and 18-CA were 4, 8, and 12, respectively. The total contraction duration was calculated as follows:

$$Total\ contraction\ duration = ROM/V_{CA} \times n$$

where $ROM$ is the range of motion of the knee extensions performed during each CA (i.e., 90˚), $V_{CA}$ is the angular velocity of the knee extensions (i.e., 60˚/s), and $n$ is the number of knee extensions performed during each CA.

## Statistical analysis

Descriptive data are presented as mean ± standard deviation. A two-tailed, independent $t$-test was used for comparisons between the stronger and weaker individual groups. Pearson's correlation coefficients ($r$) were calculated and classified as small ($0.1 \leq r < 0.3$), moderate ($0.3 \leq r < 0.5$), and large ($r \geq 0.5$) [29]. One-way analysis of variance (ANOVA) with repeated measures was performed to compare the voluntary knee extension torques produced during the last three knee extensions of the task-specific warm-up session and during the pre-test knee extension to determine whether the task-specific warm-up was completed. For %PAP, a two-way ANOVA (time [$Pre_{before}$, $Pre_{after}$, Post-10 s, Post-1 min, Post-4 min, Post-7 min, Post-10 min] × protocol [6-CA, 12-CA, 18-CA]) with repeated measures was used. Similarly, for % PAPE, two-way ANOVA (time [Pre, Post-10 s, Post-1 min, Post-4 min, Post-7 min, Post-10 min] × protocol [6-CA, 12-CA, 18-CA]) with repeated measures was used. When a significant interaction was found, the Bonferroni post-hoc test was used to examine the time-course changes in each CA protocol. Effect sizes were estimated by calculating the partial eta-squared ($\eta^2$) values (small: 0.01–0.059, moderate: 0.06–0.137, and large: > 0.138). For pairwise comparisons, effect size was determined by Cohen's $d$ (small: > 0.2, moderate: > 0.5, large: > 0.8) [29]. Statistical analyses were performed using statistical software (SPSS Statistics 27; IBM Japan, Tokyo, Japan). The significance level for all comparisons was set at $p < 0.05$.

## Results

### Comparison of stronger and weaker individuals and the relationship between % $PAP_{MVC}$ and muscle strength

An independent $t$-test showed a significant difference between stronger and weaker individuals for both absolute ($p < 0.001$ $d = 3.378$) and relative ($p < 0.001$, $d = 3.922$) muscle strength. There was also a significant difference between stronger and weaker individuals in %$PAP_{MVC}$ ($p = 0.004$, $d = 1.698$) (Table 1). A significant relationship was found between %$PAP_{MVC}$ and the absolute ($r = 0.550$, $p = 0.008$) and relative ($r = 0.570$, $p = 0.006$) muscle strength for all the participants (Fig 2).

### Task-specific warm-up session

In the task-specific warm-up session for each protocol, the numbers of knee extensions for 6-CA, 12-CA, and 18-CA were 4.4 ± 1.7, 4.9 ± 2.7, and 8.6 ± 4.8 for stronger individuals, and 7.0 ± 4.6, 7.3 ± 4.8, and 7.8 ± 1.4 for weaker individuals, respectively. Fig 3 shows no significant

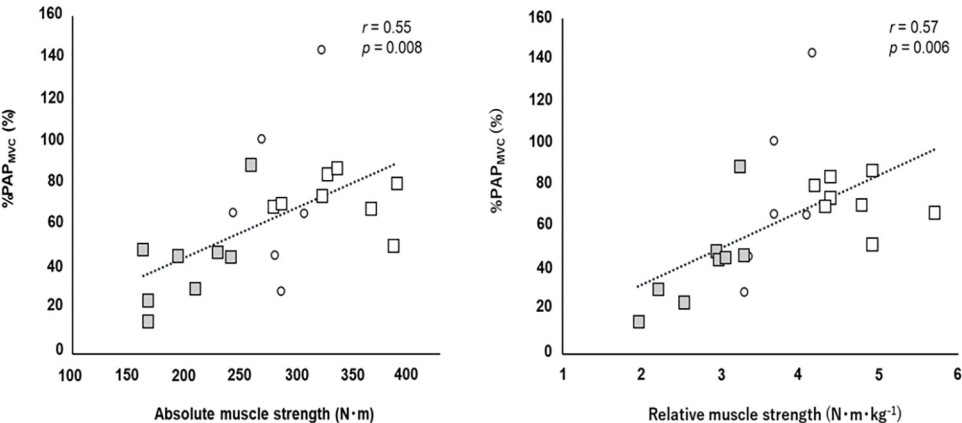

**Fig 2.** Correlation between %PAP$_{MVC}$ and absolute (*A*) and relative (*B*) muscle strength. Filled squares, open squares, and open circles represent the weaker, stronger, and intermediate levels of the relative muscle strength, respectively.

difference between the maximal voluntary knee extension torque during the last three knee extensions in the task-specific warm-up session and the maximal voluntary knee extension torque during the pre-test in each CA protocol. The fact that no significant difference was observed between the knee extension torques in the condition in which sufficient practice was conducted and in the pre-test indicates that the task-specific warm-up session was sufficiently

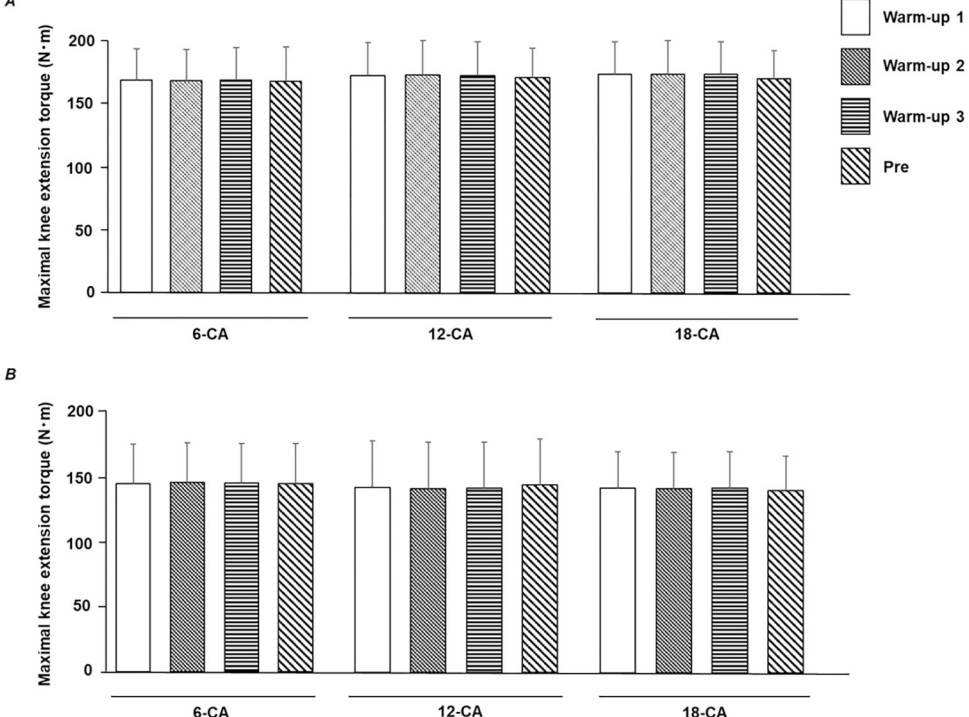

**Fig 3. The task-specific warm-up session was adequately performed.** Maximal voluntary knee extension torques produced during the last three knee extensions at the task-specific warm-up session (warm-up 1, warm-up 2, and warm-up 3) and the knee extension during the pretest for (*A*) stronger individuals and (*B*) weaker individuals. CA, conditioning activity.

completed before conducting each CA and the effects of learning and familiarization on PAPE were minimized.

## Twitch torque

For the twitch torque, the two-way ANOVA revealed that the time × protocol interaction was significant for stronger individuals ($p < 0.001$, $\eta^2 = 0.404$). Further analyses showed that in all protocols, the peak twitch torque after a set of two isokinetic knee extensions (Pre$_{after}$) was significantly potentiated (6-CA, 15.7% ± 6.3%, $p = 0.004$, $d = 1.068$, 95% CI: 10.4–21.0; 12-CA, 14.1% ± 8.4%, $p = 0.045$, $d = 0.417$, 95% CI: 7.0–21.2; 18-CA, 11.7% ± 3.6%, $p = 0.001$, $d = 1.263$, 95% CI: 8.7–14.7), indicating that two isokinetic knee extensions induce PAP. Moreover, for only 6-CA, the values of Post-10 s (29.5% ± 9.3%, $p = 0.001$, $d = 1.714$, 95% CI: 21.7–37.3) and Post-1 min (18.5% ± 6.8%, $p = 0.003$, $d = 1.101$, 95% CI: 14.8–24.2) were significantly higher than the Pre$_{before}$ value (Fig 4A). Fig 4B shows the significant interaction effect for the twitch torques for weaker individuals ($p < 0.001$, $\eta^2 = 0.369$). However, further analyses revealed no significant difference between the baseline value (Pre$_{before}$) and the values at the other time periods for each CA protocol.

## Voluntary knee extension torque

Fig 5A shows the significant interaction effect for %PAPE for stronger individuals ($p < 0.001$, $\eta^2 = 0.377$). For the 18-CA protocol, further analyses revealed that the values of Post-4 min (7.0% ± 4.5%, $p = 0.047$, $d = 0.494$, 95% CI: 3.2–10.7) and Post-7 min (8.2% ± 4.3%, $p = 0.016$,

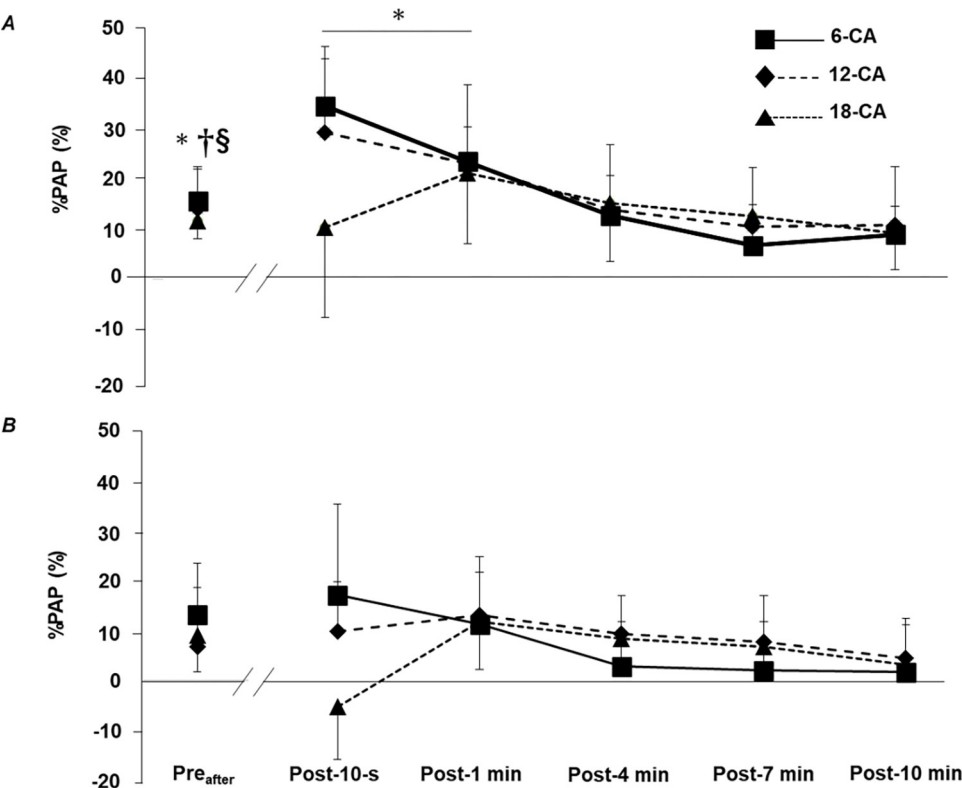

**Fig 4.** Time-course changes in %PAP after each CA for (*A*) stronger individuals and (*B*) weaker individuals. CA, conditioning activity. Values are presented as means ± standard deviation. The statistical difference is set at $p < 0.05$: difference from Pre$_{before}$ in 6-CA*, 12-CA†, and 18-CA§.

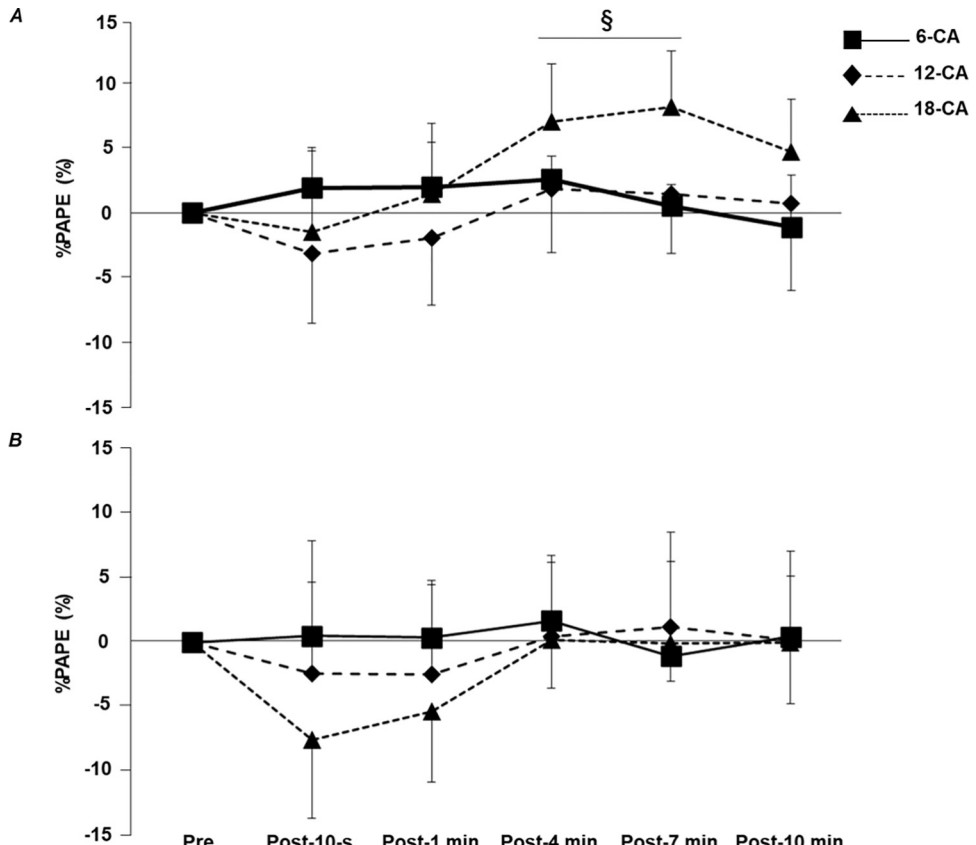

**Fig 5.** Time-course changes in %PAPE after each CA for (*A*) stronger individuals and (*B*) weaker individuals. CA, conditioning activity. Values are means ± standard deviation. The statistical difference is set at $p < 0.05$: difference from Pre in 18-CA[§].

$d = 0.568$, 95% CI: 4.6–11.8) were significantly higher than the baseline value (Pre). In contrast, there was no significant interaction for %PAPE for weaker individuals ($p = 0.057$, $\eta^2 = 0.215$). In addition, the results of %PAPE for each participant after each CA protocol are shown in Fig 6.

## Discussion

This study aimed to determine whether the optimal contraction duration of a CA to maximize PAPE differs among athletes with different muscle strength levels. The results obtained in the present study suggest that the strategies for maximizing PAPE may differ depending on the strength level of an individual.

In the present study, we observed that voluntary performance improved among the strong individuals only in the 18-CA protocol (Fig 5A), and when the results of PAPE were examined for each individual, most participants (i.e., individuals numbered 1, 3, 4, 5, 6, and 8) enhanced their voluntary performance after the 18-CA protocol more than after the other two protocols (Fig 6). These facts suggest that the CA with a relatively long contraction duration was optimal for stronger individuals. However, it should be noted that PAP had already disappeared at the time when PAPE was observed in the 18-CA protocol (i.e., Post-4 min, Post-7 min; Fig 4A), which indicates that PAP was not involved in the mechanism of PAPE in the 18-CA protocol. Previous studies have also confirmed the presence of PAPE when PAP has already disappeared

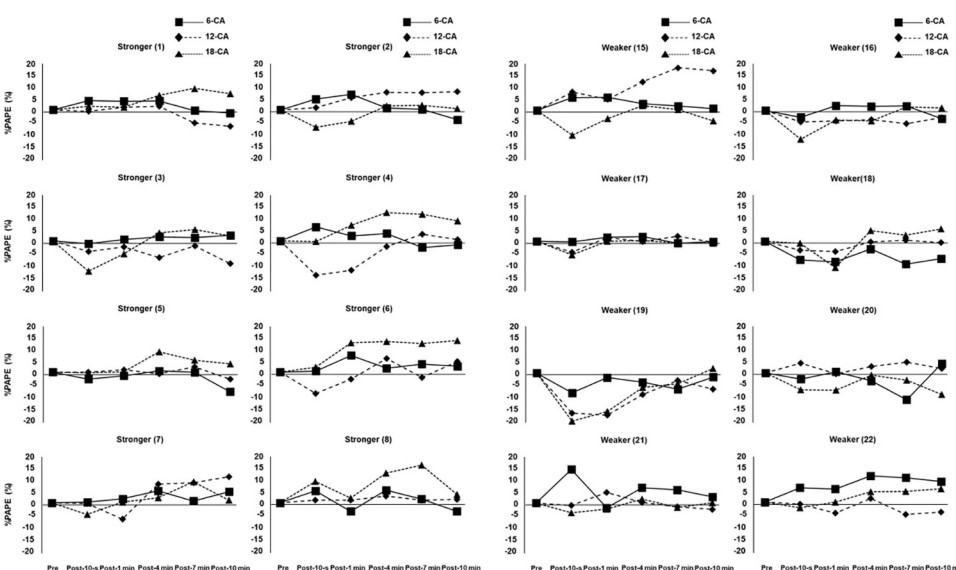

**Fig 6. Time-course changes in %PAPE after each CA for each individual.** The number in parentheses indicates how high the individual's relative muscle strength was from the top among all participants. CA, conditioning activity.

[30, 31]. Prieske et al. [30] reported that the jump performance was significantly improved after the CA, but PAP was not observed at that time and concluded that PAP was not directly linked to the voluntary performance enhancements after CA. Moreover, Thomas et al. [31] observed a significant increase in the countermovement jump height 8 min after performing the squat exercise, but PAP was no longer present at that time. Blazevich et al. [32] indicated that factors other than PAP, such as an increase in the muscle temperature, a change in the central drive capacity, and the learning and familiarization effect, may contribute to acute voluntary performance enhancement. De Ruiter and De Haan [33] reported that parameters such as maximal power production improved as muscle temperature increased. This is due to improved ATP turnover rate by facilitation of temperature-dependent myosin ATPase activity, which ultimately increases muscle shorting velocity [34, 35]. In addition, the increase in muscle temperature is thought to decrease the viscous resistance of muscles and joints and increase the nerve conduction rate [36]. Therefore, it is possible that increasing the muscle temperature associated with a CA would temporally enhance the subsequent voluntary performance. Furthermore, González-Alonso et al. [37] showed that the quadriceps muscle temperature continued to increase during a 3 min voluntary knee extension exercise. Based on this study, it is likely that the 18-CA protocol, which had the longest total contraction duration, increased the muscle temperature more than that observed with the other two protocols. Therefore, the fact that PAPE was observed only in the 18-CA protocol for stronger individuals may have been related to the difference in the degree of increase in the muscle temperature. In addition, Folland et al. [38] confirmed that the H-wave amplitude, normalized by the maximal amplitude of the M-wave, increased 5–11 min after the isometric knee extension exercise. This result indicates that a CA has the potential to increase the excitability of the motoneuron pool. Therefore, in the present study, it is possible that the CA increased the excitability at the spinal level, resulting in the increased recruitment of higher-order motor units. Furthermore, Macintosh et al. [23] referred to the possibility that the learning and familiarization effects regarding the task performance could affect the acute enhancement of the voluntary performance. However, in the present study, the participants completed the task-specific warm-up session to the point

where no further improvement in the voluntary knee extension torque production could be achieved. Thus, it is unlikely that the learning and familiarization effects on task performance influenced the PAPE in this study. Collectively, factors other than PAP and the learning and familiarization effects, such as the increase in the muscle temperature [32, 34–36] and change in the central drive capacity [32, 38], may be involved in the improvement of the voluntary knee extension performance observed in the 18-CA protocol for stronger individuals. However, since these factors were not investigated in our study, we cannot conclude whether these factors were involved in the potentiation of the knee extension performance in the 18-CA protocol for stronger individuals.

Notably, the group of stronger individuals showed PAP at Post-10 s and Post-1 min of the 6-CA protocol (Fig 4A), but failed to exhibit PAPE during the same time (Fig 5A). The finding regarding the absence of the potentiation of the voluntary performance despite the occurrence of the maximal twitch potentiation has been confirmed in other studies. Fukutani et al. [28] researched the effect of a 6-s isometric ankle plantar flexion exercise on PAP and PAPE and found that isometric twitch torque and isokinetic voluntary plantar flexion torque at 180°/s significantly increased by 62.7% and 6.4%, respectively, immediately after the CA. However, 1 min after the CA, although PAP remained significantly higher than that at baseline, its potentiation effect decreased to 30.3%, and the voluntary performance was no longer significantly different from that at baseline. Importantly, sensitivity to the potentiation effect of a CA differs between the twitch torque and maximal voluntary torque [4]. The $Ca^{2+}$ concentration is much higher in the maximal voluntary concentric contraction (multiple impulses) than in the twitch contraction (single impulse). In addition, the extent of increase in the torque by a CA becomes smaller when the $Ca^{2+}$ concentration is higher because, in this case, several cross-bridge attachments already exist [39]. In this context, Zimmermann et al. [40] noted that PAP must be extremely high to contribute to an increase in voluntary performance. Therefore, it is reasonable to conclude that the potentiation effect observed after 6-CA was insufficient to enhance the voluntary performance to the point where a statistical difference could be observed. Nevertheless, a previous study by Seitz et al. [14] confirmed that isokinetic knee extension exercises at 60°/s with a 6-s total contraction duration improved the maximal knee extension torque at 180°/s after the CA. Several previous studies have shown that the degree of potentiation effect of CA varies among individuals, even among subjects with similar muscle strength levels [41, 42], but the factors that contribute to this difference have not been fully elucidated even today. Therefore, the reasons for the discrepancy between the results of the present study and those of Seitz et al. [14] remain unclear. Future studies are needed to elucidate in more detail the mechanisms involved in the individual differences in the degree of PAPE.

In contrast to stronger individuals, weaker individuals failed to display PAPE in any of the CAs (Fig 5B), and thus, we could not conclude whether the CAs performed with relatively short or long total contraction durations were optimal for weaker individuals. The fact that PAPE was observed in stronger individuals but not in weaker individuals after the 18-CA protocol may be due to the difference in the fatigue resistance against a high-intensity exercise. Miyamoto et al. [17] confirmed that after a 5-s maximal isometric knee extension exercise, the dynamic maximal knee extension torque increased 1 to 3 min later but not immediately after the CA. Interestingly, however, after 12 weeks of high-intensity resistance training, the potentiation effect of the voluntary performance was observed even immediately after the CA. The main reason for the earlier realization of PAPE after the intervention is that the individuals developed resistance to fatigue through chronic high-intensity resistance training, suppressing the remarkable fatigue associated with a high-intensity CA. In other words, this result means that the participants with higher muscle strength levels have a greater fatigue resistance to a high-intensity CA, and there is a high possibility that weaker individuals had a lower fatigue

resistance to a high-intensity CA than stronger individuals in the present study. Therefore, it is likely that the significant fatigue associated with the CA was responsible for the lack of the potentiation of their voluntary performance in the 18-CA protocol for weaker individuals. In addition, contrary to our hypothesis, weaker individuals failed to enhance the voluntary performance after the CA with a relatively short total contraction duration (i.e., 6-CA). However, based on the individual data, we observed that for some participants (i.e., individuals number 15, 21, and 22), the CAs performed with short or moderate total contraction durations were suitable for maximizing PAPE (Fig 6). In comparison to a high-intensity CA with a relatively long contraction duration, a high-intensity CA with a short or moderate total contraction duration may produce less fatigue and allow the potentiating effect to dominate even in weaker individuals with a relatively low fatigue resistance. Future studies should continue to investigate the optimal long total contraction duration required for weaker individuals to maximize PAPE. Furthermore, the enhancement in the maximal twitch torque was also absent in all protocols for weaker individuals. Hamada et al. [1] discovered that the magnitude of PAP depends on the muscle fiber type distribution, and participants with larger fast-twitch fibers evoked higher PAP. Considering that stronger individuals tend to have higher fast-twitch fiber content [19, 20], they can trigger higher PAP through CA implementation. Fig 2 indicates that the higher the relative muscle strength, the larger the magnitude of PAP. Therefore, it is possible that the weaker individual group, composed of participants with low muscle strength levels, did not significantly increase PAP due to the reduced number of fast-twitch fibers.

The present study has some limitations. First, the athletic characteristics of the American football players who participated in this study may have influenced the results of this study. Specifically, since most of the game time in American football consists of intermittent high-intensity exercise, the participants in this study could have had higher fast-twitch fiber content, which is associated with the degree of PAP, than the athletes who require more endurance capacity (e.g., long-distance runners). Therefore, further studies should be conducted to determine whether similar results can be obtained in athletes with a high endurance capacity. Second, because the nature of isokinetic actions is different from that of isotonic actions, it may be difficult to directly apply the results to movements in sports and training. Further studies using isotonic exercise are required to apply the results of this study to the field. Fourth, the measures related to PAPE mechanisms that could help explain the mechanisms of the increase in voluntary performance observed after the 18-CA protocol, such as the central drive capacity, muscle temperature, and water content [32], were not examined in this study. Future studies should include these measures after a high-intensity CA with a long contraction duration. Lastly, the experimental conditions employed in this study did not allow us to determine whether the CAs performed with relatively short or long contraction durations are optimal for individuals with low muscle strength levels. Therefore, we hope that future studies with modification to the experimental conditions of this study (e.g., muscle contraction types, total contraction durations) will be conducted.

## Conclusion

The present study revealed that stronger individuals were able to enhance their voluntary performance only after CA with a long contraction duration (i.e., 18-CA), whereas weaker individuals could not enhance their voluntary performance in any CA. The findings suggest that the CA with a relatively long contraction duration is optimal for maximizing PAPE in individuals with high muscle strength. In contrast, we could not conclude whether the CAs performed with relatively short or long contraction durations are optimal for individuals with low muscle strength. Based on the results of this study, it is suggested that coaches need to determine the

strategies to maximize PAPE while considering the individual's strength levels. Specifically, athletes with high relative muscle strength levels may likely prefer to select a CA performed with a relatively long contraction duration to induce the potentiation of the voluntary performance sufficiently. However, in this case, as PAP is not expected to be a primary mechanism for improving voluntary performance, CA focused on enhancing PAP may not always be effective to maximize PAPE. In contrast, athletes with low relative muscle strength may need to complete chronic high-intensity resistance training as a first step to utilizing PAPE. This may lead to the enhancement of fatigue resistance against high-intensity exercise, which is thought to contribute to the realization of PAPE. To improve the practical use of the current study, future studies should examine whether the results observed in this study are reproduced when isotonic exercises are employed.

## Acknowledgments

The authors would like to thank all participants for their cooperation and Editage (www. editage.jp) for the English language review.

## Author Contributions

**Conceptualization:** Kaito Nakata.

**Data curation:** Kaito Nakata.

**Formal analysis:** Kaito Nakata.

**Investigation:** Kaito Nakata.

**Methodology:** Kaito Nakata.

**Project administration:** Kaito Nakata.

**Supervision:** Kaito Nakata.

**Validation:** Kaito Nakata.

**Writing – original draft:** Kaito Nakata.

**Writing – review & editing:** Kaito Nakata, Takaaki Mishima.

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
