## [Decision Letter · Decision Letter 0]

18 May 2022

PONE-D-22-10542The acute effects of knee extension exercises with different contraction durations on the subsequent maximal knee extension torque for athletes with different strength levelsPLOS ONE

Dear Dr. Nakata,

Thank you for submitting your manuscript to PLOS ONE. After careful consideration, we feel that it has merit but does not fully meet PLOS ONE’s publication criteria as it currently stands. Therefore, we invite you to submit a revised version of the manuscript that addresses the points raised during the review process. You will see that the the reviewers have made several comments, towards both some minor issues (e.g. use of abbreviations, clarity) as well as some more fundamental aspects (e.g. practical application, clarification on some methods used and rationale). Please consider the comments carefully and address them accordingly.  In addition, please consider the following:-Clarify whether the values in brackets after ICC are range of confidence intervals and at which level (e.g., 90%, 95%)-Please explain the advantages of the regression-discontinuity design used as opposed to a correlation between strength scores and PAP/PAPE achieved.

We look forward to receiving your revised manuscript.

Kind regards,

Dr Theodoros M. Bampouras

Academic Editor

PLOS ONE

Journal Requirements:

Reviewers' comments:

Reviewer's Responses to Questions

**Comments to the Author**

1. Is the manuscript technically sound, and do the data support the conclusions?

Reviewer #1: Partly

Reviewer #2: Yes

2. Has the statistical analysis been performed appropriately and rigorously? 

Reviewer #1: I Don't Know

Reviewer #2: Yes

3. Have the authors made all data underlying the findings in their manuscript fully available?

Reviewer #1: Yes

Reviewer #2: No

4. Is the manuscript presented in an intelligible fashion and written in standard English?

Reviewer #1: Yes

Reviewer #2: Yes

5. Review Comments to the Author

Reviewer #1: General

This is an interesting study which seeks to examine the optimal contraction duration for maximising voluntary performance and whether this varies depending on the strength level of an individual. There are some suggestions for improving the introduction and methods in particular and some questions that require consideration around the procedures and analyses.

Introduction

General – there are a number of abbreviations introduced and used in this section which impact on its readability. Please consider whether all of these (e.g. SI and WI) are necessary.

General – this is a rather long (6 paragraphs) section in which the rationale for the investigation is built but if often theoretical in nature. Please consider highlighting the practical application(s) of the research questions that you pose for the reader to fully appreciate the need for the study.

Line 56 – please provide a supporting citation for this given the ‘factual’ statement.

Methods

Line 114 – please refer the reader to table 1 for details of participant characteristics.

Line 115 – what is meant by regular resistance training (i.e. structured vs recreational, frequency)?

Line 128 – why was this method of normalising strength levels preferred over other options?

Line 131 – was 24 hours a sufficient recovery period considering the demands of the study?

Line 135 – details of the CAs are provided (i.e. joint velocity) but it is not yet clear what activities were being used for the CA.

Line 137-140 – the repeatability information may be better placed in the description of the protocols and not in the study design sub-section.

Line 141 – elaborate on the sufficiency of 10 minutes rest given pre-testing activities that may have taken place. What is meant by ‘rest’ – stationary, seated?

Line 143-149 – again, these details may be better placed in the description of the protocols and not in the study design sub-section.

Line 146 – how did ‘alignment’ take place?

Line 149 – on what basis was a cut-off frequency of 12 Hz selected?

Line 158 – please justify your choice of instruction given the variability in peak torque and rate of force development that has been shown to occur with different instructions.

Lines 234-249 – the statistical analysis needs to be explained in a way that is clearly related to the research questions. e.g. why were Pearson’s correlations used? Did the use of independent t-tests to examine difference between SI and WI not depend on the outcomes of the ANOVA considering the study design used?

Results

Line 254 – consider using ‘independent’ t-test rather than ‘unpaired’

Table 1 – height is not presented in metres. MVC/Weight is actually body mass (kg).

Discussion

Line 357-359 – include supporting citations

Line 362 – reword for clarity

Line 385 – the presumption here is somewhat speculative. Please adapt that language to indicate that this may have been a plausible explanation.

Line 421 – are the authors confident that the findings do not reflect the study being underpowered given the limitations with sample size?

Reviewer #2: General comments:

This study set out to compare the impacts of conditioning activity of varying durations on subsequent muscle performance, with additional comparisons between stronger and weaker individuals within the same cohort. Overall, this is a well-designed experiment with some interesting findings. I believe this study is certainly publishable, however, I do have several specific (minor) comments below that I feel could be addressed before publication of this manuscript. The comments below are generally line-specific, but I do have one general observation that the authors might want to consider:

There are several “unconventional” abbreviations in this manuscript that might take the reader a while to become familiarised with. Because of this, I found parts of the introduction difficult to follow and felt the need to re-read sections a couple of times to understand the point. Perhaps the authors could consider removing one or two of the non-essential, and less frequently-used, abbreviations or even re-structure sentences/paragraphs in a way that make them more readable.

This, along with addressing other comments below, would likely improve the overall quality of the manuscript.

Finally, I cannot see where to access the raw data for this project. Although I don’t think it impacted my ability to review the manuscript, I know it is a requirement of the journal.

Specific comments:

Introduction

Line 52-55: In which situations do coaches use CA? A couple of contextual examples here would be helpful, maybe just in parentheses.

Line 95: Change to “…resulting in a higher level of PAPE.”

Methods

Lines 146-149: At what sampling rate was the torque signals collected at?

Line 149: What type of filter was used on these data, a low-pass Butterworth filter or something else? Also, what was the rationale behind the 12 Hz cut-off frequency? If this was based on previous dynamometry literature, then please cite necessary articles.

Lines 165-168: What is the reasoning behind this sample size? Why did the authors specifically take the strongest and weakest eight participants, and not more (let’s say, ten?)

Line 197: “posttest” should be hyphenated. Please change all occurrences to “post-test”.

Lines 234-249: (Statistical analysis) Please cite Cohen (1988) study where relevant (lines 236 and 247).

Discussion

Lines 343-346: Could the authors expand the mechanisms that might explain how muscle temperature increases could have led to the findings of the present study?

Line 353: typographical error – change to “participants”.

Line 404: typographical error – change to “participants”.

Lines 404-410: I wonder if, at this point, the authors could discuss the concept of “high-responders” and “low-responders” in PAP-type research. This might help explain some of differences observed between individuals with similar strength capabilities.

6. PLOS authors have the option to publish the peer review history of their article (what does this mean?). If published, this will include your full peer review and any attached files.

Reviewer #1: No

Reviewer #2: No

---

## [Author Response · Author response to Decision Letter 0]

25 Jun 2022

Response to Editor

Comment 1: Clarify whether the values in brackets after ICC are range of confidence intervals and at which level (e.g., 90%, 95%)

Response: The values after ICC means a range of confidence intervals, and the level is 95%. Therefore, we have changed the following the text from (Line 225-227)

“The ICC values were 0.962 (0.897–0.989) and 0.998 (0.995–1.000) for voluntary knee extension peak torque and twitch peak torque, respectively.”

to

“The ICC values were 0.962 (95% confidence interval (CI): 0.897–0.989) and 0.998 (95% CI: 0.995–1.000) for voluntary knee extension peak torque and twitch peak torque, respectively.”

Comment 2: Please explain the advantages of the regression-discontinuity design used as opposed to a correlation between strength scores and PAP/PAPE achieved.

Response: The primary objective of this study was to determine what is the contraction duration of the CA in which athletes with high and low muscle strength levels exhibit the highest PAPE, respectively. We believe that observing the correlation between strength scores and PAPE does not allow us to clarify the primary objective of this study. Furthermore, observing the changes in PAP/PAPE over time is also an important perspective in understanding the mechanism of PAPE. In order to clarify this point of view, we believe that it is effective to employ a regression-discontinuity design and observe the changes in PAP/PAPE over time after each CA is performed.

Response to Dr Theodoros M. Bampouras (academic editor):

We thank the publisher for considering our study for publication. We have revised the manuscript based on the editor's comments.

Journal Requirements:

Response: We have revised the manuscript to conform to the format required by PLoS One.

Response: In accordance with the editor's comment, we have added the text that we provided oral and written explanations to the participants and that we obtained their written consent (Line 112-114). In addition, because this study was a low-invasive experimental method, it was not necessary to obtain parental consent for the participation of minors under the approval of the Ethics Committee. Therefore, we have added a description of it to Line 117-119.

The data on which the results of the present study are based has been uploaded to a public repository. Please check the DOI below.

Doi: 10.6084/m9.figshare.20078579

Response to Reviewer 1:

Reviewer #1: General

This is an interesting study which seeks to examine the optimal contraction duration for maximising voluntary performance and whether this varies depending on the strength level of an individual. There are some suggestions for improving the introduction and methods in particular and some questions that require consideration around the procedures and analyses.

We wish to express our appreciation to the reviewers for their insightful comments on our paper. The comments have helped us significantly improve the paper.

Introduction

General – there are a number of abbreviations introduced and used in this section which impact on its readability. Please consider whether all of these (e.g. SI and WI) are necessary.

Response: As the reviewer pointed out, the introduction was difficult to understand due to unconventional abbreviations. Therefore, throughout the text, SI, WI, and P-MRLC have been changed to stronger individuals, weaker individuals, and phosphorylation of myosin regulatory light chains, respectively.

General – this is a rather long (6 paragraphs) section in which the rationale for the investigation is built but if often theoretical in nature. Please consider highlighting the practical application(s) of the research questions that you pose for the reader to fully appreciate the need for the study.

Response: We strongly agree with the reviewer on this point. Therefore, we have removed some of the rational for the investigation in the introduction and added the following description regarding the practical examples of PAPE. (Line 47-51)

In practice, coaches can utilize PAPE through high-intensity resistance exercises as CA to enhance the performance of subsequent plyometric exercises (i.e., complex training). Alternately, they can help enhance performance by employing CA immediately prior to the actual competitive performance (e.g., 100-m performance).

In addition, due to a lack of explanation of the practical applications of this research, the following text has been added to the final paragraph. (Line 101-102)

This hypothesis, if proven, will reaffirm the importance of individualizing strategies based on the level of muscle strength.

Line 56 – please provide a supporting citation for this given the ‘factual’ statement.

Response: In accordance with the reviewer's comment, we have added the supporting citation (Line 60).

Methods

Line 114 – please refer the reader to table 1 for details of participant characteristics.

Response: In accordance with the reviewer's comment, Table 1 was referred to Line 106.

Line 115 – what is meant by regular resistance training (i.e. structured vs recreational, frequency)?

Response: In accordance with the reviewer's comment, we have added the following text to the Methods (Line 106-108).

All participants had performed traditional resistance exercises, including high-intensity squats, approximately twice a week for at least six months.

Line 128 – why was this method of normalising strength levels preferred over other options?

Response: We wish to thank the reviewer for this comment. We believe that it is important to employ relative muscle strength index to separate the subjects' muscle strength levels in order to achieve the objectives of this study. This is because the adaptations that occur with high-intensity training (high fatigue resistance to high-intensity CA, high content of fast-twitch muscle fibers), which are important in this study, are better reflected in relative muscle strength than in absolute muscle strength. If absolute muscle strength were used as the classification index, we believe it would not adequately reflect these adaptations that can occur with high-intensity training, as the effect of body size would likely influence group classification.

Line 131 – was 24 hours a sufficient recovery period considering the demands of the study?

Response: We appreciate the reviewer's comment on this point. As the reviewer pointed out, sufficient rest between sessions is necessary in this study because the effects of fatigue may affect the results. However, all the exercises used in this study were concentric contractions, and previous studies have confirmed that the muscle strength loss that occurs after concentric contractions of the knee is fully recovered the next day1). Therefore, it is expected that the minimum recovery period of 24 hours employed in this study was an adequate rest period.

1): Souron, R., Nosaka, K. & Jubeau, M. Changes in central and peripheral neuromuscular fatigue indices after concentric versus eccentric contractions of the knee extensors. Eur J Appl Physiol 118, 805–816 (2018). https://doi.org/10.1007/s00421-018-3816-0

Line 135 – details of the CAs are provided (i.e. joint velocity) but it is not yet clear what activities were being used for the CA.

Response: In accordance with the reviewer's comment, we have added the following the text regarding the details of the CAs employed in this study (Line 130-132).

“Isokinetic knee extension exercises were performed for dynamic CA, and the joint angular velocity of each CA was set at 60°/s.”

Line 137-140 – the repeatability information may be better placed in the description of the protocols and not in the study design sub-section.

Response: In accordance with the reviewer's comment, the repeatability information has been placed in the paragraph of the “Experiment session” (Line 223-227). 

Line 141 – elaborate on the sufficiency of 10 minutes rest given pre-testing activities that may have taken place. What is meant by ‘rest’ – stationary, seated?

Response: In accordance with the reviewer's comment, we have changed the following the text from (Line 134-136)

“In all sessions, the participant was required to rest for 10 min after arriving at the laboratory to eliminate the effect of PAP induced by traveling to the laboratory”

to

“In all sessions, the participant was required to limit their activities such as walking for 10 min after arriving at the laboratory and remaining at rest to eliminate the effect of PAP induced while traveling to the laboratory”

In addition, we have added the following text to explain that 10-minute rest period is sufficient to eliminate PAP (Line 136-137)

“Given that PAP declines quasi-exponentially and is negligible after about 5 minutes [1,2], this rest period is sufficient to dissipate the effects of PAP”

Line 143-149 – again, these details may be better placed in the description of the protocols and not in the study design sub-section.

Response: In accordance with the reviewer's comment, we have moved the description of dynamometer settings to Line 146-149, and the description of analog-to-digital conversion to Line 162-165.

Line 146 – how did ‘alignment’ take place?

Response: In accordance with the reviewer's comment, we have added the following the text to Line 149-151.

“Based on a previous study by Miyamoto et al. [23] the knee and hip joint flexion angles were fixed at 90° and 80°, respectively”

Line 149 – on what basis was a cut-off frequency of 12 Hz selected?

Response: The cutoff frequency of 12 Hz was determined by our own preliminary experiments. Preliminary experiments confirmed that a cutoff frequency of 12 Hz eliminates noise due to motion and avoids excessive smoothing.

Therefore, we have added the following text to Line 166-168 regarding the determination of the cutoff frequency.

“This cutoff frequency was determined in our preliminary experiments, and we have confirmed that it avoids excessive smoothing and eliminates noise due to motion.”

Line 158 – please justify your choice of instruction given the variability in peak torque and rate of force development that has been shown to occur with different instructions.

Response: In accordance with the reviewer's comment, we have added the following the text (Line 152).

“This instruction is considered favorable for measuring maximal force and power [24,25].”

Lines 234-249 – the statistical analysis needs to be explained in a way that is clearly related to the research questions. e.g. why were Pearson’s correlations used? Did the use of independent t-tests to examine difference between SI and WI not depend on the outcomes of the ANOVA considering the study design used?

Response: We thank the reviewer for this comment. Certainly, the correlation between strength level and the degree of PAP is not a result that directly confirms the objective of this study. However, if differences were observed between stronger individuals and weaker individuals in the results of PAPE after performing the CA with the same contraction duration, we thought that the results of the correlation between strength level and the degree of PAP, as shown in Figure 2, could provide further insight into the cause of these differences. Furthermore, we believe that the use of independent t-tests to identify differences in muscle strength levels between stronger and weaker individuals leads to a strong indication that the results presented in Figures 4 and 5 are the results of groups with different muscle strength levels. This is important for the purpose of this study, which is to show the results of PAPE for athletes of different strength levels.

Results

Line 254 – consider using ‘independent’ t-test rather than ‘unpaired’

Response: In accordance with the reviewer's comment, we have changed “unpaired” to “independent” (Line 239).

Table 1 – height is not presented in metres. MVC/Weight is actually body mass (kg).

Response: In accordance with the reviewer's comment, we have changed “m” to “cm” and changed “Weight” to “Body mass” (Table 1).

Discussion

Line 357-359 – include supporting citations

Response: In accordance with the reviewer's comment, references were cited in the relevant text (Line 368).

Line 362 – reword for clarity

Response: In accordance with the reviewer's comment, we have changed the following the text from (Line 373-374).

“We should note why SI was unable to improve the voluntary performance in the condition in which PAP was present in the 6-CA protocol (i.e., Post-10 s, Post-1 min, Fig 4A and Fig 5A)”

to

“Notably, the group of stronger individuals showed PAP at Post-10 s and Post-1 min of the 6-CA protocol (Figure 4A), but failed to exhibit PAPE during the same time (Figure 5A)”

Line 385 – the presumption here is somewhat speculative. Please adapt that language to indicate that this may have been a plausible explanation.

Response: In accordance with the reviewer's comment, we have revised the following text from (Line 396-399)

“Although their study did not state the years of resistance training experience, it can be presumed that the participants in their study had a longer resistance training history than those in the present study, which could explain the inconsistent results”

to

“Although their study did not state the number of years of resistance training, most of the participants in this study were college students who began resistance training in earnest during college years and most likely had fewer years of training than those in the study by Seitz et al. [14], which could explain the inconsistent results”

Line 421 – are the authors confident that the findings do not reflect the study being underpowered given the limitations with sample size?

Response: In accordance with the reviewer's comment, we believe that this study is not underpowered because we have confirmed that the power calculated from the sample size used in this study and the effect size obtained is above 0.8.

Response to Reviewer 2:

Reviewer #2: General comments:

This study set out to compare the impacts of conditioning activity of varying durations on subsequent muscle performance, with additional comparisons between stronger and weaker individuals within the same cohort. Overall, this is a well-designed experiment with some interesting findings. I believe this study is certainly publishable, however, I do have several specific (minor) comments below that I feel could be addressed before publication of this manuscript. The comments below are generally line-specific, but I do have one general observation that the authors might want to consider:

We appreciate you taking the time to offer us your comments and insights related to the paper. We found the reviewer’s feedback very constructive. The comments have helped us significantly improve the paper. We tried to be responsive to reviewer’s concerns.

There are several “unconventional” abbreviations in this manuscript that might take the reader a while to become familiarised with. Because of this, I found parts of the introduction difficult to follow and felt the need to re-read sections a couple of times to understand the point. Perhaps the authors could consider removing one or two of the non-essential, and less frequently-used, abbreviations or even re-structure sentences/paragraphs in a way that make them more readable.

This, along with addressing other comments below, would likely improve the overall quality of the manuscript.

Response: We wish to thank the reviewer for this comment. As the reviewer pointed out, the introduction was difficult to understand due to unconventional abbreviations and more theoretical content than necessary. Therefore, we have removed some abbreviations (i.e., SI→stronger individual, WI→weaker individual, P-MRLC→phosphorylation of myosin regulatory light chains). In addition, we have revised some theoretical content and reorganized the text to better convey the objectives of this study to the reader.

Finally, I cannot see where to access the raw data for this project. Although I don’t think it impacted my ability to review the manuscript, I know it is a requirement of the journal.

Response: The data on which the results of the present study are based has been uploaded to a public repository. Please check the DOI below.

Doi: 10.6084/m9.figshare.20078579

Specific comments:

Introduction

Line 52-55: In which situations do coaches use CA? A couple of contextual examples here would be helpful, maybe just in parentheses.

Response: We thank the reviewer for this comment, as supplementing reviewer’s points will help readers find the significance of this study in the field. Therefore, we have added the following text to the Introduction. (Line 47-51).

“In practice, coaches can utilize PAPE through high-intensity resistance exercises as CA to enhance the performance of subsequent plyometric exercises (i.e., complex training). Alternately, they can help enhance performance by employing CA immediately prior to the actual competitive performance (e.g., 100-m performance).”

In addition, we have added practical examples in parentheses for the areas the reviewer pointed out (Line 57-58).

Line 95: Change to “…resulting in a higher level of PAPE.”

Response: In accordance with the reviewer's comment, we have changed the following text from (Line 85)

“resulting in a greater extent of PAPE”

to

“resulting in a higher level of PAPE”.

Methods

Lines 146-149: At what sampling rate was the torque signals collected at?

Response: In accordance with the reviewer's comment, we have added a description of sampling rate to Line 168.

Line 149: What type of filter was used on these data, a low-pass Butterworth filter or something else? Also, what was the rationale behind the 12 Hz cut-off frequency? If this was based on previous dynamometry literature, then please cite necessary articles.

Response: A low-pass Butterworth filter was used for the data (Line 165-166). The cutoff frequency of 12 Hz was determined by our own preliminary experiments, not based on previous literature. In our preliminary experiments, we confirmed that a cutoff frequency of 12 Hz eliminates noise caused by exercises and avoids excessive smoothing.

Therefore, we have added the following text to Line 166-168 regarding the determination of the cutoff frequency.

“This cutoff frequency was determined in our preliminary experiments, and we have confirmed that it avoids excessive smoothing and eliminates noise due to motion.”

Lines 165-168: What is the reasoning behind this sample size? Why did the authors specifically take the strongest and weakest eight participants, and not more (let’s say, ten?)

Response: We thank the reviewer for this comment. In order to achieve the objectives of this study, we believe it is necessary to clearly separate the differences in muscle strength levels between the stronger and weaker individuals group. However, the greater the number of subjects in each group, the more likely it is that the objectives of this study will not be achieved. Furthermore, if the number of subjects in each group is set to ten in this study, one subject will be classified into the stronger individuals group in terms of “absolute muscle strength” and classified into the weaker individuals group in terms of “relative muscle strength”, which makes the discussion somewhat difficult and may mislead the reader. For this reason, we chose eight subjects for each group, recognizing the fact that the sample size was somewhat small.

Line 197: “posttest” should be hyphenated. Please change all occurrences to “post-test”.

Response: In accordance with the reviewer's comment, we have changed “posttest” to “post-test” throughout the text.

Lines 234-249: (Statistical analysis) Please cite Cohen (1988) study where relevant (lines 236 and 247).

Response: In accordance with the reviewer's comment, we have cited the Cohen (1988) study (Line 242, Line 253).

Discussion

Lines 343-346: Could the authors expand the mechanisms that might explain how muscle temperature increases could have led to the findings of the present study?

Response: In accordance with the reviewer's comment, we have added the following text regarding the mechanism by which the increase in muscle temperature associated with the implementation of CA improves voluntary movement performance (Line 346-349).

“This is due to improved ATP turnover rate by facilitation of temperature-dependent myosin ATPase activity, which ultimately increases muscle shorting velocity [32,33]. In addition, the increase in muscle temperature is thought to decrease the viscous resistance of muscles and joints and increase the nerve conduction rate [34]”

Line 353: typographical error – change to “participants”.

Response: This error has been corrected in accordance with the reviewer's comment (Line 363).

Line 404: typographical error – change to “participants”.

Response: This error has been corrected in accordance with the reviewer's comment (Line 418).

Lines 404-410: I wonder if, at this point, the authors could discuss the concept of “high-responders” and “low-responders” in PAP-type research. This might help explain some of differences observed between individuals with similar strength capabilities.

Response: We appreciate the reviewer's concerns on this point. While we found the reviewers' comments valuable, we chose not to discuss the individual differences observed between the similar muscle strength levels in this study for the following two reasons. First, to the best of our knowledge, there are very few studies that discuss differences in the degree of PAPE among the similar muscle strength levels, and we believe that discussing this point based on the limited results of this study would lead to excessive speculation. Second, since explaining individual differences within similar muscle strength levels is somewhat different from the purpose of this study, we fear that by discussing this point, the purpose of this study may not be adequately conveyed. Therefore, while we consider the reviewer's comments to be an important point for future research on PAP-type, we have chosen to avoid discussing this point in this study.

---

## [Decision Letter · Decision Letter 1]

13 Jul 2022

PONE-D-22-10542R1The acute effects of knee extension exercises with different contraction durations on the subsequent maximal knee extension torque for athletes with different strength levelsPLOS ONE

Dear Dr. Nakata,

Thank you for submitting your manuscript to PLOS ONE. After careful consideration, we feel that it has merit but does not fully meet PLOS ONE’s publication criteria as it currently stands. Therefore, we invite you to submit a revised version of the manuscript that addresses the points raised during the review process. The areas that need further work are identified below - please note that these are in addition to Reviewer 2 comments available under 'Comments to the author'.Please *justify *the selection of the regression-discontinuity design over the correlation; in other words, *why *you believe that the correlation can not meet the study's objectives, especially as you have included later and suggested it strengthens the study. Information like this will help future researchers .Please clarify in the Ethics statement that the minors that took part were able to, and gave, informed consent.Please elaborate further on the resistance training programme - what is 'traditional' resistance exercises? In what part of the season were the players in? Information like this will help future researchers as well as literature comparisons. Please *justify *the normalisation method used, rather than why normalisation was used (e.g. https://pubmed.ncbi.nlm.nih.gov/12141882/ v https://pubmed.ncbi.nlm.nih.gov/21187208/)Please provide the a-priori power calculation for the sample size (e.g. https://www.ncbi.nlm.nih.gov/pmc/articles/PMC7575068/). The use of post-hoc power analysis is incorrect, which has been previously discussed in various fields (e.g. https://gpsych.bmj.com/content/32/4/e100069, https://journals.lww.com/annalsofsurgery/Citation/2019/01000/Post_Hoc_Power_Calculation__Observing_the_Expected.48.aspx) and can result in misleading interpretation (e.g. https://pubmed.ncbi.nlm.nih.gov/32843199/, http://daniellakens.blogspot.com/2014/12/observed-power-and-what-to-do-if-your.html).Please provide some information regarding the preliminary experiments (e.g. mode, resemblance / identical to the present study). Please report height in m (I suspect there some confusion with the reviewer's initial statement re this aspect).Please revisit the explanation regarding comparison between the present and Seitz et al study. Although the training experience is indeed not provided, the average knee extension MVC torque in the present study appears to be considerably higher than the Seitz et al study. Give the premise of the present study is that stronger people have higher PAP response, the training experience not resulting in increased torque seems irrelevant to the discussion.   

We look forward to receiving your revised manuscript.

Kind regards,

Theodoros M. Bampouras

Academic Editor

PLOS ONE

Journal Requirements:

Reviewers' comments:

Reviewer's Responses to Questions

**Comments to the Author**

1. If the authors have adequately addressed your comments raised in a previous round of review and you feel that this manuscript is now acceptable for publication, you may indicate that here to bypass the “Comments to the Author” section, enter your conflict of interest statement in the “Confidential to Editor” section, and submit your "Accept" recommendation.

Reviewer #2: (No Response)

2. Is the manuscript technically sound, and do the data support the conclusions?

Reviewer #2: Yes

3. Has the statistical analysis been performed appropriately and rigorously? 

Reviewer #2: I Don't Know

4. Have the authors made all data underlying the findings in their manuscript fully available?

Reviewer #2: Yes

5. Is the manuscript presented in an intelligible fashion and written in standard English?

Reviewer #2: Yes

6. Review Comments to the Author

Reviewer #2: I would like to thank the authors for their careful responses to the comments from the Editor, my fellow reviewer, and myself. I believe the changes that have been made have improved the manuscript overall. Regarding my final initial comment about inter-individual differences, the authors’ decision not to act on this is fine; I agree with their justification. However, I do have two queries regarding their responses:

1) Regarding the choice of cut-off frequency used for filtering, this justification is insufficient. Confirming that a 12 Hz low-pass Butterworth filter “avoids excessive smoothing and eliminates noise due to motion” is vague and appears subjective. When you say that this was determined during preliminary experiments, do you mean you performed a residual analysis on pilot data? If so, please specify this. If not, and it was actually based on visual inspection of the data, then I think some explanation around what one defines as “excessive smoothing” is needed.

2) In response to my comment about sample size, the authors stated that “the greater the number of subjects in each group, the more likely it is that the objectives of this study will not be achieved”. Does this mean that you chose this sample size to allow you to confidently compare stronger and weaker individuals, or you chose this sample size to obtain the results you expected to before you conducted your experiments? It’s not clear to me, and this obviously has large implications for this point.

7. PLOS authors have the option to publish the peer review history of their article (what does this mean?). If published, this will include your full peer review and any attached files.

Reviewer #2: No

---

## [Author Response · Author response to Decision Letter 1]

6 Aug 2022

Response to Dr Theodoros M. Bampouras (academic editor):

We thank the referees for their careful reading of the manuscript and the helpful suggestions. We feel the comments have helped us significantly improve the paper. In particular, we acknowledge their highly valuable comments regarding the comparison of the present study with the study by Seitz et al.

Comment 1

Please justify the selection of the regression-discontinuity design over the correlation; in other words, why you believe that the correlation can not meet the study's objectives, especially as you have included later and suggested it strengthens the study. Information like this will help future researchers.

Response: For example, if we were to correlate the PAPE value 4 min after one of the three types of CA with muscle strength level, we might be able to show that "athletes with higher muscle strength level have higher PAPE" but we would not be able to compare the results with those of other contraction durations, and we would not be able to clarify " what is the contraction duration of the CA in which athletes with high and low muscle strength levels exhibit the highest PAPE ", which was the primary objective of this research.

In addition, the timing of PAPE expression from CA implementation is important to understand the mechanism. For example, if PAPE is triggered around 1 min after CA, it is possible that PAP is involved as one of the mechanisms (i.e., the time period when PAP still remains). On the other hand, if PAPE is triggered 7 minutes after CA, the possibility of PAP being involved in the mechanism is extremely small (i.e., the time period when PAP has already disappeared). We believe that the correlation results are not suitable for understanding the mechanism of PAPE generation because it is not possible to correspond PAPE and PAP results for each time period.

Comment 2

Please clarify in the Ethics statement that the minors that took part were able to, and gave, informed consent.

Response: In accordance with the reviewer's comment, we have changed the following the text from (p 10, Lines 130-133)

“The participants were informed verbally and in writing about the procedures involved, and possible risks and benefits of the study, for which they provided written consent before the study commenced.”

to

“The participants were informed verbally and in writing about the procedures involved, and possible risks and benefits of the study, for which written informed consent was obtained from all study participants before the study commenced.”

Comment 3

Please elaborate further on the resistance training programme - what is 'traditional' resistance exercises? In what part of the season were the players in? Information like this will help future researchers as well as literature comparisons.

Response: We thank the reviewer for this comment. As the reviewer commented, the word “traditional” was confusing, so we have changed the word to “basic” (p10, Line 122-126).

In addition, we have added a description of the more detailed training program that the subjects were conducting and changed the following text from

“All participants had performed traditional resistance exercises, including high-intensity squats, approximately twice a week for at least six months.”

to

“All participants had performed basic resistance exercises, including high-intensity bench press and squats in a session of training, approximately twice a week for at least six months. In addition, the present study was conducted in the pre-season period, and power exercises such as power cleans were also introduced into the training program.”

Comment 4

Please justify the normalisation method used, rather than why normalisation was used (e.g. https://pubmed.ncbi.nlm.nih.gov/12141882/ v https://pubmed.ncbi.nlm.nih.gov/21187208/)

Response: In accordance with the reviewer's comment, the following text has been added to Line 145-146 to justify the normalization method employed in this study.

“As body size affects the tested muscle strength, Jaric et al. [22] recommend normalization method that divides torque according to body mass.”

Comment 5

Please provide the a-priori power calculation for the sample size (e.g. https://www.ncbi.nlm.nih.gov/pmc/articles/PMC7575068/). The use of post-hoc power analysis is incorrect, which has been previously discussed in various fields (e.g. https://gpsych.bmj.com/content/32/4/e100069, https://journals.lww.com/annalsofsurgery/Citation/2019/01000/Post_Hoc_Power_Calculation__Observing_the_Expected.48.aspx) and can result in misleading interpretation (e.g. https://pubmed.ncbi.nlm.nih.gov/32843199/, http://daniellakens.blogspot.com/2014/12/observed-power-and-what-to-do-if-your.html).

Response: We thank the Reviewer for this comment. In accordance with the reviewer's comment, we determined the sample size based on a-priori power calculation (medium effect size 0.25, an alpha level of 0.05, a power level of 0.8, and three groups), and the sample size was determined to be at least 8. Therefore, the following text has been added to p 14, Line 189-193:

“The G*Power software version 3.1 was used to determine the sample size of the stronger or weaker group. The following parameters were selected: medium effect size (f = 0.25), an alpha level of .05, a power level of 0.8. The sample size for both the stronger and weaker groups was determined to be at least 8.

Furthermore, the limitation on sample size in the discussion has been removed because it did not seem to fit the revised context. (p 36)

Comment 6

Please provide some information regarding the preliminary experiments (e.g. mode, resemblance / identical to the present study). 

Response: In accordance with the reviewer's comment, we have provided a brief description of the preliminary experiment by adding the following text (p 15, Line 198-200):

“Specifically, it was determined by confirming that 3 standard deviation of the noise was no greater than the threshold of 0.5 N for the onset of contraction of isometric knee extension [27].”

Comment 7

Please report height in m (I suspect there some confusion with the reviewer's initial statement re this aspect).

Response: In accordance with the reviewer's comment, we have changed “cm” to “m”. (p 23)

Comment 8

Please revisit the explanation regarding comparison between the present and Seitz et al study. Although the training experience is indeed not provided, the average knee extension MVC torque in the present study appears to be considerably higher than the Seitz et al study. Give the premise of the present study is that stronger people have higher PAP response, the training experience not resulting in increased torque seems irrelevant to the discussion.

Response: We strongly appreciate the Reviewer's comment on this point. As the reviewer commented, it certainly appears that the participants in the present study may be stronger than the participants in Seitz et al.’s study. Thus, it is possible that our original speculation was incorrect.

The reason for this discrepancy in results is not clear, but other studies have reported large differences in potentiation effects between subjects, even at similar muscle strength levels. Therefore, it is possible that factors other than muscle strength caused the discrepancy in results between experiments, but it is unclear what these factors might be.

The discrepancies observed in the present study need to be examined in more detail in the future. Therefore, we have added the following the text (Line 456-462): 

“Several previous studies have shown that the degree of potentiation effect of CA varies among individuals, even among subjects with similar muscle strength levels [41,42], but the factors that contribute to this difference have not been fully elucidated even today. Therefore, the reasons for the discrepancy between the results of the present study and those of Seitz et al. [14] remain unclear. Future studies are needed to elucidate in more detail the mechanisms involved in the individual differences in the degree of PAPE.”

Response to Reviewer 2:

Reviewer #2:

I would like to thank the authors for their careful responses to the comments from the Editor, my fellow reviewer, and myself. I believe the changes that have been made have improved the manuscript overall. Regarding my final initial comment about inter-individual differences, the authors’ decision not to act on this is fine; I agree with their justification. However, I do have two queries regarding their responses:

Again, we express our strong appreciation to the reviewers for their insightful comments on our paper. We have incorporated changes that reflect the detailed suggestions you have graciously provided. We also hope that our edits and the responses we provide below satisfactorily address all the issues and concerns you have noted.

1) Regarding the choice of cut-off frequency used for filtering, this justification is insufficient. Confirming that a 12 Hz low-pass Butterworth filter “avoids excessive smoothing and eliminates noise due to motion” is vague and appears subjective. When you say that this was determined during preliminary experiments, do you mean you performed a residual analysis on pilot data? If so, please specify this. If not, and it was actually based on visual inspection of the data, then I think some explanation around what one defines as “excessive smoothing” is needed.

Response: We appreciate the Reviewer's comment on this point. The 12 Hz determined as the cutoff frequency was based on visual inspection of the data. Specifically, we chose 12 Hz so that 3 standard deviation (SD) of the noise was no greater than the threshold of 0.5 N for contraction onset. The 3SD argument is advocated in a 2016 paper from Maffiuletti et al1. We also believe that this value is reasonable given the mode of exercise used in the present study. The isometric knee extension and isokinetic knee extension with an angular velocity of 180° take about 250-400 ms and 250 ms to reach maximal muscle force from the contraction onset, respectively. 1/0.25 = 4 Hz, therefore, we need to set a sample frequency of 8 Hz to measure 4 Hz (Nyquist frequency), and we added 50% to lose as little data as possible. That makes 12 Hz. That can be our definition of excessive smoothing.

Therefore, we have changed the following the text from (Line 196-200)

“This cutoff frequency was determined in our preliminary experiments, and we have confirmed that it avoids excessive smoothing and eliminates noise due to motion”

to

“This cutoff frequency was determined based on visual inspection in our preliminary experiments. Specifically, it was determined by confirming that 3 standard deviation of the noise was no greater than the threshold of 0.5 N for the contraction onset of isometric knee extension [27].”

1. Maffiuletti NA, Aagaard P, Blazevich AJ, Folland J, Tillin N, Duchateau J. Rate of force development: physiological and methodological considerations. Eur J Appl Physiol. 2016;116(6):1091-1116. doi:10.1007/s00421-016-3346-6

2. Aagaard P. Training-induced changes in neural function. Exerc Sport Sci Rev. 2003;31(2):61-67. doi:10.1097/00003677-200304000-00002

3. Oliveira AS, Corvino RB, Gonçalves M, Caputo F, Denadai BS. Effects of a single habituation session on neuromuscular isokinetic profile at different movement velocities. Eur J Appl Physiol. 2010;110(6):1127-1133. doi:10.1007/s00421-010-1599-z

2) In response to my comment about sample size, the authors stated that “the greater the number of subjects in each group, the more likely it is that the objectives of this study will not be achieved”. Does this mean that you chose this sample size to allow you to confidently compare stronger and weaker individuals, or you chose this sample size to obtain the results you expected to before you conducted your experiments? It’s not clear to me, and this obviously has large implications for this point.

Response: We thank the reviewer for this comment. The sample sizes for the stronger and weaker individuals selected for this study were determined prior to conducting the experiment in order to obtain our expected results. We believe that a sample size of 8 for each of the stronger and weaker individuals from the 22 participants in this study is preferable in terms of maintaining power while clearly separating muscle strength levels.

In addition, the following statement regarding the determination of the sample size for stronger or weaker individuals using the G*Power software version 3.1 has been added to Line 189-193

“The G*Power software version 3.1 was used to determine the sample size of the stronger or weaker group. The following parameters were selected: medium effect size (f = 0.25), an alpha level of .05, a power level of 0.8. The sample size for both the stronger and weaker groups was determined to be at least 8.”

Furthermore, the limitation on sample size in the discussion has been removed because it did not seem to fit the revised context (p 36).

Again, thank you for giving us the opportunity to strengthen our manuscript with your valuable comments and queries. We have worked hard to incorporate your feedback and hope that these revisions persuade you to accept our submission.

---

## [Decision Letter · Decision Letter 2]

17 Oct 2022

The acute effects of knee extension exercises with different contraction durations on the subsequent maximal knee extension torque among athletes with different strength levels

PONE-D-22-10542R2

Dear Dr. Nakata,

We’re pleased to inform you that your manuscript has been judged scientifically suitable for publication and will be formally accepted for publication once it meets all outstanding technical requirements.

Kind regards,

Theodoros M. Bampouras

Academic Editor

PLOS ONE

Additional Editor Comments (optional):

Reviewers' comments:

Reviewer's Responses to Questions

**Comments to the Author**

1. If the authors have adequately addressed your comments raised in a previous round of review and you feel that this manuscript is now acceptable for publication, you may indicate that here to bypass the “Comments to the Author” section, enter your conflict of interest statement in the “Confidential to Editor” section, and submit your "Accept" recommendation.

Reviewer #2: All comments have been addressed

Reviewer #3: (No Response)

2. Is the manuscript technically sound, and do the data support the conclusions?

Reviewer #2: Yes

Reviewer #3: Yes

3. Has the statistical analysis been performed appropriately and rigorously? 

Reviewer #2: Yes

Reviewer #3: Yes

4. Have the authors made all data underlying the findings in their manuscript fully available?

Reviewer #2: Yes

Reviewer #3: Yes

5. Is the manuscript presented in an intelligible fashion and written in standard English?

Reviewer #2: Yes

Reviewer #3: Yes

6. Review Comments to the Author

Reviewer #2: The authors have now responded to my comments satisfactorily. I thank them again for their efforts here.

Reviewer #3: (No Response)

7. PLOS authors have the option to publish the peer review history of their article (what does this mean?). If published, this will include your full peer review and any attached files.

Reviewer #2: No

Reviewer #3: No

---

## [Editor Report · Acceptance letter]

19 Oct 2022

PONE-D-22-10542R2 

The acute effects of knee extension exercises with different contraction durations on the subsequent maximal knee extension torque among athletes with different strength levels 

Dear Dr. Nakata:

I'm pleased to inform you that your manuscript has been deemed suitable for publication in PLOS ONE. Congratulations! Your manuscript is now with our production department. 

Kind regards, 

on behalf of

Dr. Theodoros M. Bampouras 

Academic Editor

PLOS ONE